# Unlocking the Power of Light on the Skin: A Comprehensive Review on Photobiomodulation

**DOI:** 10.3390/ijms25084483

**Published:** 2024-04-19

**Authors:** Maria Luisa Hernández-Bule, Jorge Naharro-Rodríguez, Stefano Bacci, Montserrat Fernández-Guarino

**Affiliations:** 1Instituto Ramón y Cajal de Investigación Sanitaria (IRYCIS), 28034 Madrid, Spain; mluisa.hernandez@hrc.es; 2Dermatology Department, Ramon y Cajal University Hospital, 28034 Madrid, Spain; jorgenrmed@gmail.com; 3Research Unit of Histology and Embriology, Department of Biology, University of Florence, 50139 Florence, Italy; stefano.bacci@unifi.it

**Keywords:** photobiomodulation, dermatology, laser, photodynamic therapy, LED

## Abstract

Photobiomodulation (PBM) is a procedure that uses light to modulate cellular functions and biological processes. Over the past decades, PBM has gained considerable attention for its potential in various medical applications due to its non-invasive nature and minimal side effects. We conducted a narrative review including articles about photobiomodulation, LED light therapy or low-level laser therapy and their applications on dermatology published over the last 6 years, encompassing research studies, clinical trials, and technological developments. This review highlights the mechanisms of action underlying PBM, including the interaction with cellular chromophores and the activation of intracellular signaling pathways. The evidence from clinical trials and experimental studies to evaluate the efficacy of PBM in clinical practice is summarized with a special emphasis on dermatology. Furthermore, advancements in PBM technology, such as novel light sources and treatment protocols, are discussed in the context of optimizing therapeutic outcomes and improving patient care. This narrative review underscores the promising role of PBM as a non-invasive therapeutic approach with broad clinical applicability. Despite the need for further research to develop standard protocols, PBM holds great potential for addressing a wide range of medical conditions and enhancing patient outcomes in modern healthcare practice.

## 1. Introduction

### 1.1. Current Background of Photobiomodulation in Dermatology

Photobiomodulation (PBM), formerly known as low-level laser light therapy (LLLT), is a safe phototherapy technique that uses wavelengths of the visible light spectrum which includes red light (RL, 620–700 nm) and near-infrared (NIR, 700–1440 nm) [1]. This treatment modality has been used in dermatology, both in clinical settings and at home. PBM involves the use of various light sources [2], including low-level lasers (LLL) and light-emitting diodes (LED), to deliver therapeutic light [3]. Reviews on the matter have already been conducted [1,3,4,5,6,7].

In recent years, the application of PBM has expanded rapidly, driven by a growing body of research supporting its efficacy in treating various dermatologic conditions. From acne and wound healing to skin rejuvenation and scar reduction, PBM offers a non-invasive and promising alternative to conventional therapies [4].

Probably one of the best-studied applications of PBM in dermatology is in conditions secondary to oncology therapies, such as radiation dermatitis or mucositis. PBM has been proven to decrease the severity, progressive worsening, and pain of radiation dermatitis through multiple controlled trials, achieving a level of evidence of IA for this application [4,8]. In this regard, two recent meta-analyses reached the conclusion that there is evidence to support PBM as a protective treatment against severe radiation dermatitis [9,10]. PBM has also been used to treat postmastectomy lymphedema as it can reduce limb volume/circumference and extracellular fluid index [11].

PBM has also been used to treat hair disorders, including non-scarring alopecia such as male and female androgenic alopecia, alopecia areata, as well as scarring alopecia like frontal fibrosing alopecia and lichen planopilaris. It has been proven to prolong the anagen phase of the hair cycle through the release of paracrine growth factors from dermal papilla cells [12]. Increased hair density and length have repeatedly been observed in large clinical trials [13,14].

Regarding acne, there is evidence that PBM can reduce skin sebum and transepidermal water loss [15]. Moreover, PBM light may be absorbed by local flora such as *P. acnes*, leading to the destruction of this bacteria [16]. Several studies, including clinical trials, have demonstrated significant reductions in the number of inflammatory lesions [17]. PBM use in skin rejuvenation is justified by the apparent remodeling effect through the production of type 1 and type 3 collagen and elastin [18]. Finally, PBM may play a role in improving healing outcomes of wounds and scars. It has been shown to influence all of the wound healing phases as well as in reducing inflammation [19].

One of the key mechanisms of PBM involves the absorption of photons by endogenous photoreceptors, including mitochondrial cytochrome C oxidase (COX) (1). Compared to other treatment modalities with light, PBM offers several advantages. PBM is non-invasive, cost-effective [20], and convenient for patients, with a very favorable safety profile [21]. Additionally, PBM can be used as an alternative or in combination with traditional pharmacological therapies [22]. Despite PBM’s promising prospects, standardized recommendations for PBM treatments across skin conditions and different skin types are lacking in the literature.

### 1.2. Importance of Light Therapy in Dermatology Practice

Dermatologists play a crucial role in understanding and delivering PBM effectively since they have traditionally used light sources to effectively treat skin conditions for decades, including lasers [23] and phototherapy [24]. PBM shares biological, physical, and physiological mechanisms in common with laser therapies, more specifically chromophores [25], wavelengths, and penetration depth [26]. Phototherapy, similarly to PBM, aims for modulatory effects on the skin without inducing thermal damage or inflammation [27].

At the heart of light therapy lies a deep understanding of the intricate interplay between light and skin biology. Dermatologists wield this knowledge to address a spectrum of skin diseases that will be further developed later. By applying different wavelengths, dermatologists can tailor treatments to target specific cellular processes, modulate inflammation, promote collagen synthesis, and accelerate tissue repair, among others.

One of the most compelling aspects of light therapy is its versatility. Whether delivered via lasers or light-emitting diodes (LEDs), light therapy can be customized to suit individual patient needs and skin types. This flexibility allows dermatologists to optimize treatment outcomes while minimizing adverse effects, thus maximizing patient satisfaction.

In addition to its therapeutic efficacy, light therapy boasts an impressive safety profile, making it suitable for patients of all ages and skin tones. Unlike most pharmacological agents, light therapy carries minimal risk of systemic side effects, making it an attractive option for pregnant women, children, and individuals with sensitive skin; this aspect will be further discussed later.

Furthermore, the advent of home-based light therapy devices has democratized access to this transformative modality, empowering patients to take control of their skin health from the comfort of their own homes. This accessibility fosters greater patient engagement and adherence to treatment regimens, ultimately enhancing treatment outcomes and overall quality of life.

Therefore, we believe that dermatologists should lead the advances in PBM treatments and help provide the necessary scientific evidence. Nevertheless, contemporary medicine, with the goal of developing and optimizing treatment approaches, must not disregard basic research and biological principles investigated in the laboratory. Therefore, the progress of PBM development and its future clinical applications must be developed in close collaboration with basic research investigators.

### 1.3. Review Objectives

Clinical trials about photobiomodulation LED light therapy or low-level laser therapy and their applications in the dermatology field published over the last 6 years were reviewed. Thus, this paper aims to provide a comprehensive overview of the current role of photobiomodulation in dermatology with a special interest in the molecular mechanism of action and the recently discovered or poorly analyzed applications (skin rejuvenation, wound healing and scars) helping the reader to obtain a perspective of the immediate future steps of this promising therapy. Through a collaborative effort between dermatologists specializing in light-based therapies and basic researchers delving into photobiology, we endeavor to illuminate the potential of PBM in promoting skin health and fostering the development of novel therapeutic paradigms. By bridging the gap between scientific inquiry and clinical application, we aspire to catalyze the advancement of photobiomodulation as a cornerstone of modern dermatologic practice.

## 2. Basic Molecular Mechanisms of Action

Endre Mester, a Hungarian physician, first created the idea of photobiomodulation (PBM) in 1967 while researching the effects of laser light exposure on the growth of cancer cells in a mouse model. Mester discovered an unanticipated acceleration of hair regeneration during this investigation [28]. Naturally, since its inception, light therapy has undergone tremendous changes and has grown in scope. PBM has demonstrated efficacious outcomes in the management of non-healing wounds, scars, ulcers, musculoskeletal disorders, persistent pain, analgesia, and immunological regulation [29].

Increasingly, research indicates that particular electromagnetic radiation wavelengths, from visible to infrared, can produce photophysical and photochemical effects that can alter important biological processes in a variety of eukaryotic organs, including humans. Hence, non-ionizing light sources in the visible and infrared range, such as broadband lights, low-level lasers, and Light-Emitting Diodes (LEDs), are used in treatment to activate endogenous chromophores and stimulate biological functions in a non-thermal and, most importantly, non-cytotoxic manner [30,31].

The main characteristic of PBM is the direct interaction of continuous wave light at specific wavelengths directly with the tissue, i.e., with the endogenous chromophores.

However, due to a lack of understanding of the photo-physics and radiometric parameters that affect PBM’s accuracy and reliability, as well as a lack of knowledge about how PBM achieves its molecular effects, there is still disagreement about its practical application.

### 2.1. Blue LED Therapy

Blue LED technology was created in the early 1990s and has since found many uses in the biochemical and biological domains [32,33]. In fact, blue LED technology offers the possibility of a straightforward and affordable source for focusing on specific biological components, given that a number of biological molecules and chromophores exhibit a high absorption rate in the UV/blue area of the spectrum [34]. One use for it is the process of wound healing.

Blue light is absorbed mainly by the heme group, which can be found in the hemoglobin and in the cytochromes. These widely distributed biological elements have the capacity to trigger one or more intracellular signaling pathways following radiation exposure, which can alter the healing process [35]. Protoporphyrin IX is a chromophore that increases the sensitivity of cytochrome C and cytochrome C oxidase to blue light in the mitochondrial electron transport chain [35,36,37,38].

Cytochrome C and cytochrome C oxidase, once activated by blue light, strengthen the process of cellular respiration by interacting with the final two complexes of the chain and adjusting the ATP production [38,39]. For this reason, mitochondria represent a target organelle for blue light radiation. Mitochondria are involved in redox signaling and in maintaining the balance of reactive oxygen species (ROS), which is essential for several vital functions such as calcium homeostasis. In a preliminary study, Magni et al. [40] demonstrated that under blue light exposure, ROS are stimulated dose-dependently and that mitochondria are subject to morphological changes.

According to André-Lévigne et al. [41] and Dunnill et al. [42], flavins, which stimulate the synthesis of ROS and play a role in the signal transduction mechanism in numerous cellular pathways involved in tissue repair, can also be used to explain the therapeutic action of blue light. The shift from this phase to the proliferative one is facilitated by the modulation of ROS, which causes a controlled increase in inflammatory functions [43,44]. The macrophage phenotypic transition from M1 to M2 may also be responsible for this [29,30].

Compared to wounds that are not treated with blue light, wounds treated with it exhibit a quicker healing process as well as improved dermal collagen deposition and morphology [34,45,46,47,48,49]. Additionally, the modulation of the inflammatory response is better in wounds that have been treated. His process can be attributed to photochemical effects: nitric oxide release and fibroblast activation brought on by blue light promote re-epithelialization [35,50,51]. According to Fraccalvieri et al. [39] and Orlandi et al. [52], blue light helps injured tissue by regulating its energy supply, which is particularly important during the phases of proliferation and remodeling. It also establishes tissue repair, with a decrease in scar tissue and the likelihood of developing keloids, as well as the stimulation of angiogenesis [52,53].

In animal models of chronic ulcers, blue light treatment has also been shown to have a proangiogenic effect; this suggests that blue light therapy may be helpful throughout the whole tissue repair process [54]. In addition, blue light has been shown to have anti-inflammatory cytokine release, bacterial load reduction, and granulation stimulation [40,50,51,55]. The safety and efficacy of blue light in treating skin lesions, Inflammatory acne [56,57], burns [51], psoriasis [58,59,60,61], eczema [62], and diabetic ulcers are among the conditions that blue light is reported to be safe and effective for treating. Despite the evidence presented above, relatively few methodologically rigorous experiences have been conducted in daily clinical practice focused on chronic wounds [39].

In an in vitro research context, Rossi et al. [46] investigated the impact of blue LED light on the proliferation and metabolism of human fibroblasts derived from healthy skin cocultured with keratinocytes. As described in their article, the authors suggested using blue LED light to modulate the metabolism and growth of human fibroblasts.

Furthermore, human fibroblasts isolated from keloids and perilesional tissues were subjected to blue LED light irradiation, which was examined by Magni et al. [45]. The authors utilized various experimental techniques to demonstrate that blue LED light can modulate cell proliferation and metabolism in a dose-dependent manner and that these effects persist for at least 48 h after treatment.

Moreover, in keloid-derived fibroblasts and perilesional fibroblasts, the highest radiation doses decrease cell viability 24 and 48 h after treatment, respectively. In order to treat hypertrophic scars and keloids, the authors concluded that blue light irradiation was a novel and minimally invasive treatment option.

### 2.2. Photodynamic Therapy

Today, photodynamic therapy (PDT), which was first developed by Von Toppeiner and Jesionek in partnership [63], is extensively utilized to treat a wide range of illnesses [64]. The application of this therapy in dermatology spans the spectrum from treating bacterial, fungal, viral, immunological, or inflammatory illnesses to managing chronic wounds, including photorejuvenation in cosmetology [65].

The application of photosensitizers activated by a specific wavelength of light energy is used in this therapy: the topical use of 5 Aminolaevulinic Acid (ALA) has represented a breakthrough in PDT in the dermatological field because it is easily absorbed by the skin [66,67].

The molecular mechanism of action of PDT is complex. When triplet oxygen (3O_2_) is present, the photo-sensitizer can enter the tissue. The main purpose of PDT is to have a selective photokilling effect on a pathologic target while promoting healing in healthy tissue. In order to reach this goal, a photosensitizer is used and included in the target tissue. The photosensitizer has an absorption peak at a specific wavelength: when it is exposed to a light source (usually a laser) emitting at this wavelength, it generates Reactive Oxygen Species (ROS) that are very unstable. The generation of ROS can vary depending on the energy input and the characteristics of the photosensitizers [54]. While less noticeable levels of ROS may promote tissue proliferation and/or regeneration, high levels may cause a photokilling effect. This approach has been usually proposed in the treatment of cancerous cells or in the inactivation of multidrug-resistant pathogens [68].

PDT is used in medicine to treat a wide range of pathologies, both oncological and non-oncological [69,70]. As almost all the therapies are based on the use of light sources, its advantages include minimal invasiveness, ease of use in an outpatient setting, and a strong track record of short- and long-term safety.

Intravenous injection of photo-sensitizer can cause damage at the vascular level. Hypoxic damage can be beneficial in treating neoplasms, but it can also be detrimental in treating chronic non-neoplastic wounds because it exacerbates the hypoxic state, which is a major factor in noxa.

Without a single, perfect photo-sensitizer, the selection process should focus on molecules that have a proven track record of improving skin ulcer healing and can be applied topically. Two reviews [68,71] have recently examined this vast topic. Photo-sensitizers for chronic ulcers have been tested using various chemical categories, primarily in preclinical settings and a few pilot studies. These chemical categories include phenothiazines (methylene blue and toluidine blue), xanthenic dyes (rose bengal), and tetrapyrrolic macrocycles and analogs (porphyrins, chlorines, and phthalocyanines).

The selection of the light source is also very important. In clinical practice, PDT is performed using laser or LED (Light Emitting Diodes) sources. Lasers are strictly monochromatic, thus enabling an excellent matching with the photosensitizer absorption curve, high fluences and a spatially narrow beam. On the other hand, LED sources usually present a large emission spectrum but are more affordable, and their wide angular emission can cover larger tissue areas. The wavelength controls how well light can pass through tissue. Specifically, lengths between 600 and 800 nm are thought to have sufficient skin penetration to be utilized in clinical settings. Fundamentally, red is the color that penetrates the skin the most, followed by green and blue [72].

Given that the technique relies on a tissue’s interaction with light energy, it is clear that the effectiveness is directly correlated with the total amount of energy applied per treated area’s volume. A number of parameters, including power, irradiance, energy density, irradiation time, and release of light mode (simple or fractional), can be used to express this [73]. Energy density, which is expressed as J/cm and is derived from the measurement of time (in seconds) and irradiance (in W/cm^2^), is the most widely used format for reporting a PDT treatment schedule [74].

According to research on wound healing, particularly chronic wounds, PDT can cause an acute inflammatory response that is primarily related to immune system activation [75].

This is supported by the description of how PDT not only causes new fibroblasts (effector cells) to diversify but also fosters close relationships between these cell types and mast cells, which are positive for TNF-α and Fibroblast Growth Factor (FGF) in their granules. This was reported by Corsi et al. [76].

Thus, these results support the hypothesis that mast cells could transmit signals for the same fibroblast recruitment and differentiation following therapy [76]. When it comes to mast cells, they proliferate and degranulate during the course of treatment. Their increase could have resulted from nearby cells migrating, precursors already present in the tissue differentiating, or precursors entering the tissue and differentiating into mast cells.

As a result, these cell types would be both attracted and stimulated to release their granules into the dermis in reaction to the therapy. During therapy, the venules, or the vessels of the sub-papillary plexus, appear to be a preferred location for cell infiltration and clustering [76].

The presence of TNF-alpha, GM-CSF, and TGF-β in mast cells after PDT treatment provides additional evidence of immune system activation. TNF-alpha plays a crucial role in differentiating dendritic cells, including plasmacytoid cells, which interact with regulatory T-type lymphocytes. GM-CSF is also involved in this process. TGF-beta is essential for the differentiation of macrophages.

According to Grandi et al. [67], there is no doubt that the induction of TGF-beta is connected to the subsequent decrease in wound volume. In fact, TGF-beta seems to affect the epithelial-mesenchymal transition—which permits keratinocyte migration from the margins toward the wound bed—at different phases of ulcer healing. Additionally, this cytokine can stimulate myofibroblast differentiation as a component of the processes observed in scar reshaping [77].

Additional research has demonstrated that PDT significantly affects neutrophil activation, which may account for the rise in pro-inflammatory cytokines following treatment. Lipid mediators are produced in tandem with the acute phase of inflammation resolution and the restoration of tissue homeostasis). These mediators have been linked to anti-inflammatory and immunomodulatory properties, such as the inhibition of leukocyte chemotaxis, the blockade of TNF-alpha and IL-6 production, and the induction of increased IL-10 expression [78,79,80,81]. As a result, we can draw the conclusion that PDT significantly affects the immune system, having both immunostimulatory and immunosuppressive effects. It also likely influences the type of cell death that is induced.

According to Steinmann [82], the nervous system has the ability to control the immune system’s activity; ulcer healing is another example of this close relationship. Actually, results from experiments indicate that neurogenic stimuli significantly impact wound healing following injury and that delayed wound healing following skin nerve resection is seen in animal models [83,84].

Studies have demonstrated that following PDT treatment, there is an increase in neuronal populations belonging to the autonomous nervous system, which is found in the dermis. These neuronal populations contain the typical nerve mediators involved in ulcer healing (CGRP, NGF, NKA, NPY, SP, PGP 9.5, and VIP). Additionally, after a single irradiation, the proportion of mast cells that contain and secrete VIP and NGF rises. Given that mast cell degranulation is stimulated by both VIP and NGF, these results appear to be consistent with the previously documented rise in mast cell degranulation index following PDT treatment, indicating that neurogenic stimuli may play a role in this phenomenon.

In light of this, we can presume that an increase in NGF and VIP release following therapy results in mast cell activity and that these mediators can activate dermal neurons and nerve fibers [81,85,86]. Corsi et al. [76] suggest that increased TGF-beta, cellular infiltrate response, and increased ECM secretion by fibroblasts may all be related to the activation of nerve fibers.

Due to its gaseous nature and relatively short half-life, NO is the smallest known signaling molecule that can cross membranes freely. It has recently been added to the list of mediators involved in wound healing [87]. In fact, the presence of bacterial antigens, apoptotic bodies, or inflammatory cytokines increases the expression of the enzyme, which suggests this molecule is derived from the NOS enzyme complex, where the inducible isoform is overregulated during stressful situations.

The inflammatory phase of wound repair, which is characterized by the promotion of vasodilation and antibacterial activity, has been theorized to be facilitated by iNOS [81,88,89]. Experimental preliminary results show increased expression of iNOS in chronic wounds treated with photodynamic therapy. In contrast to granulocytes to M2-type macrophages, vessels, and neurons where iNOS expression increases, mast cells have a higher degranulation index and contain iNOS; however, the proportion of these cells containing this mediator decreases following treatment [90].

However, research is currently being done in the lab to determine how different cell types respond to PDT in terms of iNOS secretion and subsequent wound healing.

### 2.3. LED vs. Low-Level Laser Light Therapy Comparison

The treatment known as PBM has been labeled by various terms. The term “photobiomodulation” is more widely accepted among authors as it refers to the general mechanism, whereas using the term LLLT may confuse the reader into thinking that PBM can only be done with lasers [91].

PBM is used to refer to the interaction of light sources with a target modulatory action on specific biological reactions or pathways. The term LLLT arises from the discovery of the photobiomodulatory effects of lasers on the periphery of treated lesions. Despite the fact that this term has been widely used, PBM effects on the skin can be obtained not only by applying a laser at low energies with that intention. LED lights or non-coherent sources, without seeking selective photothermolysis, can also be used for PBM [92] (Figure 1).

Consequently, there are two main ways to apply PBM in dermatology. One method involves using LED lights, while the other utilizes low-dose lasers below the selected target threshold. Both light sources differ in some key aspects. Laser light is coherent and exact, whereas LED light is non-coherent in a bandwidth of 1–2 nm [91]. The application methods also vary. Despite using low doses [93], lasers deliver high energy in a short time, resulting in short sessions administered by an expert dermatologist [94]. Laser devices are more expensive. LEDs, on the other hand, are simple, more affordable devices that do not require specialized handling. LEDs apply energy over a longer duration compared to lasers [92]. For PBM applications, both devices require repeated sessions, with protocols not clearly established, typically ranging from once per week to multiple times per week [2]. Most of the published works on PBM utilize lasers as light sources, representing up to 90 percent of the more than 3000 published works [91]. This could lead to the wrong assumption that PBM is only achieved with lasers. However, numerous studies currently support the idea that LEDs are another valid option for applying PBM. Table 1 shows the differences between laser and LED devices when used as light sources in PBM.

The illumination in PBM differs when using lasers or LEDs. With lasers, the area to be treated is covered with spot overlap using the handpiece [103], whereas with LEDs, the device is simply fixed at a certain static distance (1–20 cm) or applying slight movements over the treated area [104]. Additionally, the illumination differs whether an LED lamp with a single bulb or when multiple LED arrays are used. LED light delivery can be continuous or pulsed at different frequencies.

The use of light in a no-thermal effect is supported by the photon’s absorption of the cells’ receptors. The three main chromophores in the skin are melanin in the epidermis, hemoglobin in the dermis, and water in all the skin, with longer wavelengths achieving deeper penetration (Figure 2) (1). Red light (RL) targets melanin and hemoglobin, whereas NIR light targets water in the deeper layers of the epidermis [105] (Figure 3). Blue light (BL 400–500 nm) has been included in some devices, but it is considered very close to ultraviolet light with deleterious effects on the skin and without modulatory effects [106].

## 3. Current Applications in Dermatology

### 3.1. Acne Treatment

#### 3.1.1. Effectiveness of Photobiomodulation in Acne

When the existing evidence on PBM in acne is reviewed, the term PBM becomes once again frequently confused with low-power lasers or LLLT. Thus, clinical trials on lasers, intense pulsed light at low doses, and LEDs for the treatment of acne have been published. Additionally, studies on photodynamic therapy in acne overlap with PBM, as LED lights are generally used as light sources [107]. Numerous studies with LED and BL are found for acne treatment, especially with home devices [108,109]. BL, due to its demonstrated harmful effects, will not be the focus of this review [106]. Remarkably, most laser studies in acne are focused on the treatment of scars, where the laser is considered one of the first-line treatments [23]. These studies should not be confused with those focusing on low-dose lasers and PBM.

Clinical trials of acne treated with low-power lasers have generally been focused on reducing inflammation and improving healing. For this purpose, the selected lasers have been the KTP (535 nm) [110], pulsed dye laser (PDL 585–595 nm) [111,112,113], Diode (1450 nm) [114] and Nd:YAG laser (1054) [113] since they operate in the correct wavelengths (Figure 3).

Multiple sessions were applied throughout two to three months at low doses of laser in acne, looking for the effects of PBM from the laser. In a study, the KTP laser has been shown to be effective in reducing inflammatory lesions and erythema in acne [110]. PDL is one of the most studied lasers in acne due to PDL’s ability to reduce erythema and inflammation. In a randomized, split-face, blind study compared to no treatment, no reduction in acne severity or erythema grade was found over two months of treatment [111]. In another study, Park et al. compared PDL with a non-ablative 1550 nm laser and found that both devices significantly reduced acne lesions with slight superiority of the 1550 nm laser [112]. PDL laser in acne was also compared with Nd:YAG 1065 in the treatment of inflammatory acne lesions, and both showed efficacy in treating acne lesions, with both treatments being effective [113]. Diode laser 1450 nm has been shown to be capable of reducing inflammatory acne lesions by 62%, and when combined with BL, it seems to decrease seborrhea [114]. All these findings lead to the belief that lasers used at low doses within the appropriate spectrum range can reduce inflammation in acne.

Remarkably, there are few clinical trials regarding LED lights and PBM in acne. All the studies again applied repeated sessions, once or twice a week, for at least two months. In an interesting study, narrow-band ultraviolet B at low doses was compared with treatment using 630 nm red light and oral erythromycin as a control [115]. A much more significant improvement was found with narrow-band ultraviolet B. However, the use of UVB treatments must be restricted and strictly controlled by dermatologists due to UVB’s potential to induce carcinogenesis. Another study demonstrated that there were no differences between alternating RL and BL or applying them in the same session and that one session per week is sufficient in inflammatory acne with LED BL-RL [116]. When photodynamic therapy (PDT) was compared with intense pulsed light (IPL) and the combination of RL-BL, PDT demonstrated higher effectiveness. PDT achieved a 92% reduction in acne inflammatory lesions compared to 58% with IPL and 44% with the combination of RL-BL [117].

The combination of RL-BL at-home devices was effective, specifically in reducing inflammatory acne lesions by 77%. In this study, biopsies of the lesions were taken, showing a decrease in inflammatory acne lesions, inflammatory cell infiltration, and sebaceous gland size [118]. As an interesting combination treatment, a study showed that treatment with NIR-L prior to PDT increases its effectiveness in PDT acne treatment [107]. RL at a dose of 6 to 9 J/cm^2^ in combination with treatment with isotretinoin in acne improves skin dryness and tolerance to treatment [119]. RL-BL could be better at reducing inflammatory lesions than 10% salicylic acid peels repeated for acne treatment [120].

From all these clinical trials, it can be inferred that the effectiveness of LED light treatment in acne is moderate for inflammatory lesions, and its role would be limited to isolated treatments or in combination for moderate forms of acne or those not eligible for medical treatments.

#### 3.1.2. Underlying Mechanisms in Acne Treatment

Acne is a skin pathology caused mainly by the bacteria *Cutibacterium acnes*. These bacteria are found in the sebaceous glands of all individuals and is part of the skin microbiome, helping to maintain its balance. When this imbalance is lost, some strains of *Cutibacterium acnes* disappear while others become predominant. This selective infection causes dysbiosis or imbalance of the microbiome [121]. In turn, other bacteria from the staphylococcus family (mainly *Staphylococcus epidermidis* and also *Staphylococcus aureus*) proliferate, accentuating the imbalance of the microbiome. Thus, the imbalance of the skin microbiome is accompanied by the formation of biofilms that isolate bacteria from the outside, allowing them to continue growing and making them more resistant to antibacterial treatments. On the other hand, *Cutibacterium acnes* exerts an action on the cells responsible for sebum production, which leads to the overproduction of sebum [122]. This dysbiosis is the main cause of inflammatory acne and its chronic nature. Common treatments involve the use of topical agents such as antibiotics or retinoids or systemic drugs such as retinoids, antibiotics and hormonal agents [121]. However, these therapeutic strategies may develop side effects such as antibiotic resistance, alteration of the microbiome, or present limited efficacy. Therefore, PBM represents an interesting therapeutic alternative for the treatment of acne.

On the one hand, photobiomodulation has an antimicrobial effect by inhibiting the proliferation of bacteria responsible for acne, particularly *C. acnes*. The mechanism of such an antimicrobial effect is due to the absorption of light by porphyrins, a byproduct of its metabolism which functions as an endogenous photosensitizer. This process triggers a photochemical reaction that generates reactive free radicals and singlet oxygen forms [122]. It has been described in *C. acnes* that these porphyrins are activated with certain wavelengths, especially in the UVA or blue light spectrum. Thus, Cho et al. [123] observed the photoinactivation effect of light irradiation on *C. acnes* with wavelengths of 370 nm, 385 nm, 395 nm, 405 nm, and 470 nm. However, they did not observe any photoinactivation effect on the bacteria at wavelengths of 505 nm, 590 nm, 630 nm or 880 nm. UVA or blue light exposure to *C. acnes* leads to the generation of reactive oxygen species (ROS), which have potent bactericidal effects, effectively reducing the population of acne-causing bacteria on the skin.

Another important mechanism of phototherapy involves the modulation of sebaceous gland activity. Excessive sebum production is a hallmark feature of acne, and photobiomodulation has been shown to regulate the function of sebaceous glands and reduce keratosis of hair follicles. In an in vitro study, Li et al. [119] obtained normalization of keratinization within sebaceous glands after exposure to red LEDs. Jung et al. [124] also demonstrated that red LEDs (630 nm) reduce lipid production in a clinical trial.

Furthermore, the colonization and proliferation of *C. acnes* are known to be crucial for the development of inflammation. The peptide cell wall of *C. acnes* initiates the release of cytokines such as IL-α, IL-1β, IL-8, and TNF-α by monocytes, which triggers the inflammatory response in the skin [122]. Photobiomodulation has been shown to have cytokine-mediated anti-inflammatory effects. Irradiation with specific light wavelengths, such as red and near-infrared, can reduce the production of pro-inflammatory cytokines and inhibit the activity of inflammatory mediators. Thus, Li et al. [119] demonstrated the anti-inflammatory effect of the method used by reducing the level of interleukin IL-α. This helps relieve redness and swelling associated with acne lesions. Also, a low dose of blue light LED exposure (415 + 470 nm) reduced the production of interleukin 8 in patients with acne, and after this exposure, the microcysts, pustules and inflammatory nodules almost disappeared, with a lasting effect [125]. Ash et al. [108] analyzed the effect of blue LED light on reducing inflammatory lesions in 41 patients with mild to moderate acne vulgaris. All subjects in the treatment cohort achieved a reduction in inflammatory lesion counts after 12 weeks, compared to the control group that used only a facial cleanser containing salicylic, glycolic, and lactic acid. In another study conducted with blue LED (15 min and 4 weeks of follow-up), short-term irradiation significantly reduced inflammatory acne compared to topical treatment with benzoyl peroxide (5%) twice a day [126]. Regarding red LED acne treatment, one study revealed a reduction in inflammatory acne (87.7%) after a 12-week follow-up treatment in 14 participants [127].

Thus, these cellular mechanisms of photobiomodulation in the treatment of acne, such as the modulation of inflammation, antimicrobial effects, and the regulation of the activity of the sebaceous glands, together with the non-thermal and non-invasive nature of photobiomodulation, offer a novel, safe and well-tolerated strategy to address this common dermatological problem.

### 3.2. Photorejuvenation

#### 3.2.1. Reduction of Fine Lines and Wrinkles

Table 1 summarizes the clinical trials studying PBM in the treatment of wrinkles [95,96,97,98,99,100,101,102,128].

Most studies focus on RL, NIR, and amber light (AL), a light of 590 nm close to red light [95,96,97,98,99,100,101,102,128]. RL has shown improvement in wrinkles and signs of photodamage when applied alone [36] in repeated sessions. Histologically, a decrease in aging damage, such as an increase in type I collagen and a slight decrease in metalloproteinases, was observed. The combination of RL and NIR [96,97,99] was evaluated, and a significant improvement in wrinkles, smoothness, and skin firmness was observed. RL, NIR, and their combination [98] were compared, and RL was found to be more effective in reducing blemishes and dark stains since NIR showed higher effectiveness in improving skin elasticity and wrinkles. These authors recommended using the combination of both RL and NIR when treating photoaging. The combination of RL with white light was evaluated in two studies, and it was found that both produced significant improvement in wrinkles without finding differences between the application of both types of wavelengths [100,101]. RL was compared with amber light (AL) of 590 nm for the treatment of periorbital wrinkles, and both lights achieved a decrease in wrinkles around the eyes, with slightly better results with RL. Some studies only included women [101,128] or found better results in women [98]. All the studies applied various sessions per week, from one session per day to two sessions weekly; follow-ups were conducted for 9 to 13 weeks and even up to 6 months. The findings were conclusive with the target light used, which is melanin in RL and water in NIR, which has a higher wavelength and penetrates deeper into the skin. NIR was able to reduce deeper wrinkles but was not as effective in reducing pigment as RL. All clinical trials reported that PBM was a safe, athermal treatment with no side effects. PBM is safe and can be applied as a treatment for wrinkles and photoaging. Nevertheless, clinical studies are not numerous, and protocols are varied, making it difficult to draw conclusions. More studies in the field are needed.

#### 3.2.2. Stimulation of Collagen Production

Collagen is a protein produced by dermal fibroblasts that is crucial for the skin as it provides structural support. However, due to aging, this protein tends to decrease, which leads to a loss of elasticity, expression lines and skin wrinkles. There are various non-pharmacological therapies that promote collagen formation by fibroblasts. One of them is phototherapy, which is applied through the non-invasive cosmetic procedure known as photorejuvenation. Phototherapy takes advantage of the effects of cellular photobiomodulation to stimulate collagen production, which improves the general appearance of the skin. This technique has become an option chosen by people seeking skin rejuvenation but who do not wish to receive invasive procedures [129].

The wavelengths of light used in photorejuvenation are carefully selected to stimulate collagen production while minimizing damage to adjacent tissue. To date, there is no agreement on the optimal wavelengths of the radiation used, but in the vast majority of cases, wavelengths from 630 nm (red light) to 950 nm (near-infrared light) are used since they are well absorbed by the chromophores of the skin and penetrate to the deepest layers where collagen production occurs.

Photorejuvenation acts through two complementary and sequential cellular mechanisms: (1) selective photothermolysis and (2) the induction of wound healing responses. Selective photothermolysis takes advantage of the absorption capacity of specific wavelengths by endogenous skin chromophores such as hemoglobin. This absorption induces a thermal increase in the dermis, which generates controlled damage. As a consequence, the tissue processes of dermal wound repair are triggered [130]. An example of this type of PBM treatment is the non-ablative fractional diode laser (NFDL) system. This device uses fractional photothermolysis to rejuvenate the skin, using two infrared wavelengths, 1440 nm and 1927 nm. Water molecules absorb infrared energy from the NFDL system, resulting in skin rejuvenation and treatment of dyschromia in the skin of color, with a reduced risk of adverse effects seen with other fractional lasers. Furthermore, the photothermolysis generated facilitates the administration of small molecular weight compounds, such as L-ascorbic acid, through the skin without compromising the skin barrier function [129]. Once photothermolysis is generated, the wound regeneration process begins (See above for details). In this process, fibroblasts migrate to the injured area and begin their regeneration through the formation of an extracellular matrix composed of collagen components, among others. Additionally, fibroblasts also proliferate and release growth factors and cytokines that further stimulate collagen formation. Thus, in an in vitro study carried out by Barolet et al. [131], exposure of human fibroblasts to 660 nm LED increased the secretion of procollagen and decreased the expression of MMPs. On the other hand, in human keratinocytes, a decrease in the expression of MMP-9 was observed after the application of the red laser (635 nm), which favored the conservation and production of new ECM [132]. Positive conclusions were also reached in another recent in vitro study [18] in which human fibroblasts were treated with red and infrared light for 10 min each day at an intensity of 0.3 J/cm^2^. This treatment induced greater expression of collagen and elastin. It has been speculated that this effect on collagen described by Wen-Hwa Li may be due to the change in the mitochondrial membrane potential generated by light exposure, which would promote the stimulation of some signaling pathways and the activation of transcription factors, mainly AP-1 and NF-kB. This activation would increase the expression of genes related to collagen synthesis, anti-inflammatory signaling, cell migration and proliferation, as well as the production of anti-apoptotic proteins and antioxidant enzymes [133]. Also, in a study conducted in mice, Neves et al. [134] combined a topical hydrogel rich in Lycium barbarum polysaccharides and PBM (red laser; 660 nm and 40 J/cm^2^) to evaluate whether isolated or combined treatments would reduce photodamage on the skin generated by UVR. The results showed that the combined treatment inhibited UVR-induced skin thickening, decreased the expression of c-Fos, c-Jun and MMP-1, −2 and −9, and increased collagen I and III levels and FGF2.

In summary, PBM activates cellular mechanisms such as photothermolysis and wound healing, which generates a new extracellular matrix with components, such as new collagen fibers, elastin, and others, which contribute to the restoration of skin elasticity and a more youthful appearance. Thus, PBM alone or in combination with chemical treatments is a promising strategy for the repair of photodamaged skin and presents a potential clinical application in skin rejuvenation.

### 3.3. Wound Healing

The main role of the skin is to serve as a defensive barrier against the surroundings. Extensive damage or disease affecting significant areas of the skin might result in profound impairment or potentially fatal outcomes. However, core mechanisms have been largely preserved throughout evolution since they are essential to life [135].

Mast cells are essential for coordinating the cellular infiltration response that follows an injury, and their role in wound healing is central [136]. Changes in these processes can result in delayed healing or even failure to heal the wound [135,137,138].

The mechanisms behind the various stages of this phenomenon largely overlap in space and time. In summary:

Coagulation phase: The coagulation phase is the first stage of the hemostasis. It starts with hemorrhaging and platelet aggregation, followed by temporary vasoconstriction brought on by the release of vasoactive chemicals by injured cells. Platelets, a vital source of cytokines for leukocyte and macrophage activation, also play a role in blood clot formation. A series of biochemical events culminate in the production of an insoluble fibrin network, which is initiated by the platelet aggregation process.

Inflammatory phase: Substances released by MCs, such as histamine and serotonin, mediate the subsequent vasodilation process after blood vessels’ initial constriction. This initiates the process of diapedesis, which involves the movement of blood corpuscle components, specifically neutrophil granulocytes followed by macrophages, to be determined by increased blood flow in the wound area.

Proliferative phase: Granulation tissue forms as a result of the proliferative phase. The fibroblasts are essential at this stage because they produce the precursors of collagen, elastin, and other molecules that make up the extracellular matrix. They also play a role in controlling the migration and proliferation of the cellular agents that are involved in neo-angiogenesis and the process of re-epithelialization.

Maturation phase: Remodeling a wound may require a year or longer. Two unique processes in humans are responsible for this phenomenon: wound contraction and collagen restoration. Myofibroblasts facilitate wound contraction and the production of scars in both adult and pediatric patients. This process leads to an increase in tensile strength, which reaches around 80% of that of unwounded skin and is associated with lysyl oxidase-induced collagen crosslinking.

The intricate process of wound healing is largely regulated by molecules that are secreted at various phases of the process, including cytokines and growth factors. The modulation of this process is crucial, as any deviation can lead to impaired wound healing and the subsequent development of circumstances conducive to chronic wound formation. TNFalpha, IL6, and IL1beta are examples of pro-inflammatory cytokines that particularly work to draw inflammatory cells to the site of injury.

Different growth factors, including TGFbeta and PDFG, are secreted by the inflammatory cells at the site of the injury. These growth factors attract fibroblasts that are actively proliferating. In order to promote the growth of epithelial cells, macrophages and fibroblasts secrete FGF2 (bFGF), Keratinocyte Growth Factor (KGF), FGF7, EGF, Hepatocyte Growth Factor (HGF), TGFalpha, and Insulin-like Growth Factor (IGF) 1. VEGF and PDGF, secreted by fibroblasts, keratinocytes, and macrophages, induce activation of endothelial cells.

The expression of genes encoding different molecules, including cytokines, chemokines, and growth factors, defines the several phases of wound healing and their interconnection. Genes that promote inflammation and produce molecules such as TNFalpha, IFNgamma, or TGFbeta are activated shortly after an injury occurs.

Genes coding for molecules like VEGF, PDGF, FGF2, and MMP, which stimulate fibroblast and keratinocyte proliferation, epithelialization, angiogenesis, and the start of eventual repair, are included in the gene profile as wound healing advances. The genes that encode TGFbeta1 and MMP expression are upregulated during the remodeling phase to encourage fibroblasts’ production of collagen and the ECM’s removal during tissue resorption.

Changes in gene expression can impact the healing sequence and result in the release of factors such as chemokines, growth factors, and cytokines. This can cause chronic wounds to develop [135,137,138].

#### 3.3.1. Chronic Skin Lesions

Any skin lesion that does not heal in six to eight weeks is considered chronic, according to international literature. The inflammatory reaction in these lesions either persists over time, balancing degenerative and productive phenomena, without proceeding through the regular, systematic, and timely sequence of the reparative process, or it progresses through these phases without managing to restore the tissue’s anatomical and functional integrity [81,137,138,139,140].

With approximately 140 diseases that can potentially display this behavior and an average of six simultaneous diseases among individuals over 65 years old (with 85% of the population affected by at least one chronic disease and 30% having three or more chronic diseases), there are numerous factors that contribute to the delay in the process, leading to blockage and ultimately chronicity.

In terms of probability, this could result in 1406 different clinical scenarios. The literature has identified and documented a number of clinical scenarios and syndromes related to the etiology of skin ulcers; however, a discussion of these is outside the purview of this paper [81,137,138,139,140,141,142].

Chronic wounds are typically characterized by long-lasting and ongoing inflammation, as indicated by previous studies. In contrast, acute healing involves the resolution of the inflammatory response. Indeed, discerning whether chronic inflammation is caused by a long-term open wound and its continuous exposure to bacteria, or if it is the other way around, or even a combination of both, poses a significant challenge. The presence of specific types of immune cells can be advantageous in certain cases of chronic wounds. Typically, a significant increase in the number of natural immune cells entering chronic wounds and their continued presence is likely to hinder various healing processes [81,138,139,140]. A recurring hindrance in the healing process of several chronic wounds is the accumulation of necrotic debris in the periphery of the wound, potentially caused by the diminished ability of immune cells to engulf and remove waste material in chronic wounds. Therefore, it is common in medical practice to remove dead tissue from the wound, either by mechanical means or by employing maggots (fly larvae), in order to create a new and healthy wound. This process promotes the efficient regrowth of the outer layer of skin (re-epithelialization) [143,144].

Prolonged inflammation in ulcers causes high protease activity, which in turn causes growth factors and other molecular cues that support the reparative phase to degenerate. Moreover, the overproduction of hydrolytic enzymes and pro-inflammatory cytokines in chronic wounds inhibits the primacy of reparative processes over destructive ones [81,138,139,140].

Thus, it has been proposed that protease activity should be decreased to preserve endogenous growth factors and facilitate the regular reparative process. As a result, while proper equilibrium between the development of new tissue and its physiological destruction is essential for the normal reparative process, it has been demonstrated that in chronic skin lesions, there is a negative correlation between tissue inhibitors of matrix m (MMPs) and elevated MMP levels. This leads to altered ECM reorganization and increased degradation [81,138,140].

Keratinocytes at the edge of a chronic wound show signs of partial activation, as seen at a molecular level. This includes the increased expression of certain genes involved in cell division, such as cyclins, and the suppression of genes that regulate the cell cycle and p53. This could potentially explain the excessive growth of the epidermis observed at the edges of ulcer wounds [138,139,143]. The fibroblasts in an ulcerated wound appear to be in a state of senescence, with reduced ability to migrate [76], and they show limited response to the migratory stimulant transforming growth factor-β (TGF-β) [81]. One possible reason for the decrease in growth factor signaling and sensitivity could be the higher levels of tissue-degrading matrix metalloproteinases (MMPs) observed in chronic wound-tissue fluids compared to acute ones [67,81].

Additionally, infections are a significant and common cause of blockages in the repair process. An increase in the bacterial load prolongs the inflammatory phase, which in turn produces high levels of MMP and exacerbates the ECM’s destructive processes [138,140].

The potential uses of light in medicine, particularly in the field of dermatology and the treatment of skin malignancies, have drawn significant attention since the beginning of time [105]. In particular, two different approaches of phototherapy were proposed for the treatment of different pathologies, and in particular for promoting wound healing: photobiomodulation and photodynamic therapy. These treatments are based on the use of a monochromatic (or quasi-monochromatic light) that is absorbed by a target in the tissue: an endogenous chromophore in the case of the photobiomodulation, an exogenous chromophore in photodynamic therapy.

#### 3.3.2. Reduction of Hypertrophic Scars and Keloids

Some clinical trials focus on the study of PBM on scars, both with lasers and LED light. Lasers in scar treatment are the first line of treatment when used at standard doses, but those lasers are typically ablative, such as erbium or CO_2_, which operate with selective necrosis at the target [23]. A clinical trial in 10 patients with keloids used PDL 585 5 J/cm^2^ compared with the classic management of intralesional triamcinolone acetonide at low doses, high doses and intralesional 5-fluorouracil. Weekly treatments were applied for 8 weeks, and no significant differences were found between the four different treatment groups [128]. Asilian et al. [145] evaluated 69 patients in a 12-week double-blind study with three treatment groups: PDL at doses of 5–7 J/cm^2^, intralesional 5-fluorouracil and the combination of PDL at the same parameters with 5-fluorouracil with intralesional triamcinolone acetonide. No statistically significant differences were found, but a better aesthetic outcome and less erythema were obtained in the group treated with PDL. The same PDL doses were used in a trial of 19 patients, comparing pulse duration of 0.45 milliseconds versus 40 milliseconds, yielding better results in the short pulse duration [146]. The combination of 532 nm laser at low doses with silicone patches has also proven effective in improving hypertrophic scars in a group of 37 patients [147]. PDL 595 nm was compared with Nd:YAG 1064 nm long-pulsed at modulating doses over six sessions, and improvement in hypertrophic scars was observed. No differences were noted between both laser devices [148]. PDL 595 alone [149] and Nd:YAG 1064 alone [150] have proven effective too, when used independently. In conclusion, PDL at subpurpuric doses was the most used laser in combination with other treatments in treating hypertrophic and keloid scars to improve aesthetic outcomes and erythema, but it typically did not add benefits in terms of results. Longer wavelengths, such as 1064, were probably effective, but only two clinical trials have been published in this regard.

### 3.4. Psoriasis

PBM has been explored in psoriasis as an alternative treatment. Four clinical trials on the efficacy of PBM in psoriasis have been published [151,152,153,154], all using lasers as light source, three employed PDL [151,153,154] and one used Nd:YAG [152]. PDL has not been proven to be more effective than the combination of salicylic acid with clobetasol propionate in plaque psoriasis [151], nor superior to narrow-band ultraviolet B phototherapy [153]. Nd:YAG laser did not show efficacy [152]. A promising indication derived from a clinical trial is the use of PDL in nail psoriasis, where improvement was observed in this especially challenging location in which phototherapy or creams are ineffective due to penetration issues [154].

### 3.5. Radiation Dermatitis

PBM has been studied in radiation dermatitis as a way of relieving symptoms and reducing inflammation. PBM was first studied in a clinical trial in 2010 to prevent radiation dermatitis in breast cancer patients. The 18 patients treated with red LED did not experience a decrease in the incidence of radiation dermatitis reactions [155]. Subsequently, LED therapy was applied to 22 patients treated with RT for breast cancer and compared to controls, resulting in an improvement in radiation toxicity [156]. Moreira-Costa et al. conducted a clinical trial in which 26 patients undergoing radiotherapy for breast cancer were compared with another 26 not treated with PBM [157]. Red LED therapy was applied before and after each radiotherapy session for preventive purposes, with better results in tissue repair and inflammation reduction. Robinjs et al. published three clinical trials using 808 nm pulsed NIR PBM at a dose of 168 mW/cm^2^. In those trials, through repeated sessions, the treatment was found to reduce scaling [158] and prevent severe reactions in head and neck cancer radiotherapy treatment [8], but not in breast cancer radiotherapy [159]. In summary, the studies published on PBM in radiation dermatitis provide contradictory information regarding which type of radiation dermatitis to select for treatment and which wavelength to work with.

## 4. Future Directions

### 4.1. Technological Advances in Photobiomodulation

#### 4.1.1. Development of Newer Therapeutics by Using Alternative Wavelengths

The development of devices that apply new light modalities not previously used in photobiomodulation (PBM) is an area of ongoing research focused on optimizing the therapeutic results obtained with current devices, as well as expanding the range of applications. New research in this area seeks to advance the development of technologies that apply light radiation with wavelengths not previously used or combinations thereof, as well as the development of new light sources that allow expanding the spectrum of pathologies to be treated and reducing potential side effects of these treatments. New treatments try to expand the range of the spectrum, for example, by using light of infrared wavelengths beyond the near-infrared. These wavelengths have unique interactions with water molecules and may have specific applications for treating deeper tissues or influencing cellular processes in novel ways [160]. UV light is also being studied for its ability to modulate immune responses and promote certain cellular functions [161,162]. Results from multiple studies suggest that NB-UVB may have the potential to reduce the pathology of B cell-induced immune conditions by reducing inflammatory cell-cell communication and the production of inflammatory cytokines. Such a mechanism would possibly involve the induction of type I IFN and its associated pathways [163]. Modulation of these immunoinhibitors could play an essential role during PBM UV [164] induced systemic immunosuppression.

Another current development in the research is to delve into the applications of blue light as an antimicrobial. It has been described that bacteria (Gram-positive, Gram-negative, mycobacteria), fungi (yeasts and filamentous fungi), viruses (DNA and RNA), and parasites can be effectively destroyed by light [165,166]. Furthermore, antimicrobial efficacy appears not to be affected by microbe resistance to antibiotics, nor does it lead to resistant microbes after repeated sublethal light applications [167]. Light in the range of 400–470 nm has been described to have antimicrobial effects due to its ability to produce ROS [125,168], and although the wavelength range of 402–420 nm is the most effective, wavelengths of 455 nm and 470 nm waveforms have antimicrobial potential for some specific bacterial species (e.g., *S. aureus*) [169]. Therefore, the use of this wavelength in dermatology may be useful for wound healing treatment [170].

The combination of several wavelengths is also of great interest. This approach attempts to take advantage of the synergistic effects of different wavelengths to improve therapeutic results. For example, a novel device combining three wavelengths (1064 nm, 810 nm and 755 nm) in which the absorption and penetration properties of each were combined has been shown to be effective and safe for hair removal [171]. In another study, the efficacy of dual NIR treatment using 810 nm pulsed and 904 nm superpulsed PBM lasers for transdermal burn repair in rats was analyzed [172]. The results revealed an acceleration of burn wound healing. Noirrit-Esclassan [173] also showed efficacy in treating oral mucositis in children by applying a PBM combination of two wavelengths (635 and 815 nm).

Adapting PBM to specific clinical applications is another key objective. These investigations try to find specific wavelengths for certain pathologies, such as neurodegenerative [174,175] or musculoskeletal [176,177]. This specificity is based on the unique absorption characteristics of the target tissues, which would allow more precise and effective treatments.

Finally, the development of new light sources, such as light-emitting diodes (LED) or superluminescent diodes (SLD), may allow the development of new PBM equipment that combines specific wavelengths for certain pathologies or allow the application of new treatment protocols. These new devices will offer advantages in terms of cost, portability, and ease of integration into various treatment modalities [178,179].

#### 4.1.2. Improvements in Device Portability

In recent years, innovation in the design of PBM devices has focused on portability and the inclusion of new technologies in the equipment [180]. The improvements achieved in this field have contributed to greater accessibility and versatility of these devices, which benefits both clinical staff and patients who use them in home environments [181]. The main improvements and innovations in the design of PBM equipment are discussed below.

(1)Multimodal functionality: PBM’s newest equipment combines phototherapy with other physical therapies such as photothermal therapy, magnetic hyperthermia, cold plasma therapy, sonodynamic therapy, or radiotherapy, which completes the treatment possibilities [182].(2)Miniaturization of light sources: Advances in light-emitting diode (LED) and laser diode technologies have enabled the miniaturization of light sources. These new diodes are smaller but just as powerful, allowing the design of compact PBM devices without reducing the intensity or effectiveness of the applied light [180,183,184].(3)Portable and more flexible designs: These devices allow, on the one hand, the ability to adapt to different body contours, thus increasing the comfort of the treatments. On the other hand, as they are portable, the patient can be treated in their home environment, which avoids trips to hospitals or medical clinics [185,186,187].(4)Easier-to-use interfaces: PBM’s new equipment designs incorporate more intuitive user interfaces as well as touch screens and voice commands, making them easier to use. This ease of use, together with its portability, has allowed the patient to apply their own treatment, which provides them with greater independence and quality of life [188,189].(5)Integration with smart devices: The new devices are designed to be able to connect through wireless networks such as Bluetooth or Wi-Fi to smart devices such as mobile phones, tablets, etc. [189,190]. These connections are very useful for the user and/or patient since they allow the treatment parameters to be personalized through applications specifically designed for these devices and their real-time or remote monitoring.(6)Longer-lasting batteries: Some of the new devices have been designed to incorporate long-lasting rechargeable batteries, which do not require connection to a power source. This also improves the patient’s quality of life by allowing mobility independently of a continuous electrical connection [188,189]. In addition, being more compact, users can transport their devices easily, avoiding interruptions in their treatment.

### 4.2. Personalized Therapy in Dermatology

#### Use of Genomics and Skin Profiling for Targeted Treatments

Recent studies have suggested that the photobiostimulatory effect of PBM could influence genomic stabilization since sublethal levels of PBM radiation could activate DNA repair mechanisms. PBM could also influence telomere stabilization by modulating the mRNA expression of genes related to telomere stabilization, such as TRF1 and TRF2 [191].

On the other hand, the use of personalized medicine in dermatology involves the creation of patient-specific genetic profiles that allow the integration of their genomics with the PBM with the aim of achieving the most optimized treatment for their pathology. It has been described that personal genomic variations can affect the responses to photobiomodulation treatments. These genetic variations can influence factors such as the efficiency of mitochondrial function, antioxidant capacity, and susceptibility to inflammation. Therefore, a comprehensive analysis of skin characteristics, including moisture levels, elasticity, and pigmentation, as well as the identification of biomarkers associated with skin conditions or aging, contributes to a better understanding of the specific needs of each person. Thus, this novel discipline has significant potential to improve treatment efficacy and personalize therapeutic interventions in dermatology and skin health.

Personalized/precision strategies in dermatology are based on the identification of biomarkers that are most frequently derived from tissue transcriptional expression, genomic sequencing, or circulating cytokines of a specific pathology of interest. Based on this, atopic dermatitis and nodular prurigo may be candidate conditions for precision dermatology [192]. Recently, innovative techniques have been developed to obtain transcriptomes in skin conditions, other than biopsy and minimally invasive, to reveal different patient skin profiles. For example, methods have been developed that include applying patches to a psoriasis plaque for a few minutes to capture the epidermis/upper dermis transcriptome. Thanks to these innovative techniques, several potential biomarkers or predictors of this pathology have been found, such as biomarkers for its diagnosis such as nitric oxide synthase 2/inducible nitric oxide synthase (NOS2/iNOS), human beta-defensin-2 (hBD-2), matrix metalloproteinases 8/9 (MMP8/9), risk biomarkers for developing the pathology such as the filaggrin (FLG) gene mutation, or candidate biomarkers for monitoring the effects of treatment such as LDH, TARC, pulmonary chemokine and activated regulated (PARC), periostin, IL-22, eotaxin-1/3, and IL-8 [193].

In a recent in vitro study, Tripodi et al. [194] evaluated transcriptomic changes in human dermal fibroblasts in response to polarized PBM (P-PBM). The results showed a total of 71 differentially expressed genes (DEGs). All DEGs were found in the PBM group polarized with respect to the control group (PC). Of these 71 DEGs, 10 genes were upregulated, and 61 were downregulated. Most DEGs were related to mitochondria or extracellular matrix (ECM). The DEGs of P-PBM were almost always downregulated compared to the control groups. This may be because P-PBM treatment decreased cellular stress. Therefore, genomic analyses of the individual’s skin can detect genes related to mitochondrial function, which is fundamental for the mechanisms of PBM.

Information on the genomic profiles of the skin also makes it possible to address specific skin problems and design ad hoc PBM treatments. For example, knowledge of certain biomarkers of an individual’s genetic predisposition to inflammation could facilitate the selection of specific wavelengths or treatment durations to effectively modulate inflammatory responses. Additionally, knowledge of skin profiles and genomic information will enable the development of predictive models for treatment results. Thus, these predictive models can help professionals in selecting the most effective treatment protocols for the patient.

Clinical trials and ongoing research initiatives in personalized dermatology are still very scarce, so a large amount of study will be necessary to achieve more convenient, non-invasive and effective predictors and biomarkers to better guide personalized and precise treatment.

### 4.3. Potential in Treating Severe Skin Conditions

#### 4.3.1. Plaque Psoriasis

Psoriasis is a chronic, recurrent, immune-mediated inflammatory disease. It is associated with genetic predisposition, autoimmune disorders, psychiatry and psychological health, as well as environmental factors such as infection, stress or trauma, etc. [195]. In this pathology, the nuclear factor κB (NF-kB) pathway is activated, as well as the differentiation of T helper (Th) cells towards Th1 and/or Th17 cells. Due to this, immune cells release an excess of pro-inflammatory cytokines, among the most important IL-17, IL-21, IL-22, IL-23, and IL-26. These cytokines stimulate the proliferation of keratinocytes and increase the secretion of TNF-α and chemokines, which improve the activation of dendritic cells. This leads to the inflammation characteristic of the pathology, which is manifested by the development of thick, red, scaly spots on the surface of the skin.

PBM has emerged as a complementary and promising therapeutic strategy for the treatment of this pathology. Thus, applying PBM along with standard therapies such as topical medications or systemic treatments may offer a synergistic approach to managing psoriasis symptoms. Research in this area focuses on determining optimal wavelengths and treatment protocols for PBM in psoriasis. This technique was originally developed with the idea of using broadband ultraviolet B light (BB-UVB, 290–320 nm) for this pathology. However, later studies demonstrated the greater effectiveness of narrow-band ultraviolet B (NB-UVB, 311 nm) and even an excimer laser/lamp (308 nm) used as a monochromatic UVB source [24]. There are currently many types of phototherapies for psoriasis, including psoralen and UV-A (PUVA) (320–400 nm) and the aforementioned BB-UVB and NB-UVB. The latter has been used as first-line phototherapy for plaque psoriasis due to its better efficacy, longer remission time and fewer adverse reactions [196]. It has also been described that the combination of acitretin and NB-UVB can achieve better efficacy with fewer adverse reactions in treating plaque psoriasis [197].

On the other hand, studies that have explored the use of red and near-infrared light have shown that it can be useful for reducing psoriatic lesions and controlling abnormal skin proliferation due to the ability of PBM to accelerate wound healing and reduce inflammation. Thus, PBM has been shown to be effective in modulating the immune response and reducing the production of pro-inflammatory cytokines [198]. This anti-inflammatory response would act on the activity of immune cells such as T lymphocytes, which could improve the inflammatory response of psoriasis. Ablong et al. [199] investigated the effectiveness of the combination of 830 nm (near infrared) and 630 nm (visible red light) emitted by a light-emitting diode (LED) to treat recalcitrant psoriasis. When patients with plaque psoriasis were treated sequentially with LEDs emitting continuous 830 and 633 nm in two 20 min sessions over 4 to 5 weeks, clearance rates of 60 to 100% were achieved without significant side effects.

Given the ability of PBM to promote the production of collagen and other components of the extracellular matrix, the application of this therapy can improve the integrity of the skin barrier. Thus, this therapy can contribute to the normalization of the growth and differentiation of skin cells, potentially reducing the thickness and peeling of psoriatic plaques. A prospective randomized study comparing the effectiveness of blue light (420 and 453 nm, LED) in the treatment of psoriasis once daily for 4 weeks showed significant improvement at either wavelength [58]. For its part, light (400–480 nm) can reduce the proliferative activity of keratinocytes, modulate the immune responses of T cells and safely improve plaque psoriasis. Thus, in another study in which patients received high blue intensity (90 J/cm^2^; protocol: Every day (30 min) for 4 weeks and 3 times a week for the next 8 weeks), a significant improvement in LPSI symptoms compared to the control group [60].

Other studies have analyzed the combination of natural compounds with PBM. This is the case of the study by Niu T et al. [200], in which the effect of curcumin with blue and red LED light for the treatment of psoriasis was analyzed. This assay showed that co-treatment of curcumin and PBM downregulated the phosphorylation level of Akt and ERK, caused inhibition of NF-κB activity, and activated caspase-8/9. In a recent study, Krings et al. [201] investigated the effectiveness of blue light (453 nm, 600 mW/cm^2^, 15 min or 30 min irradiation) in the treatment of mild psoriasis vulgaris. Both treatments showed similar improvement, regardless of their duration. Additionally, PBM may be effective in treating the itching and discomfort typical of plaque psoriasis, improving the overall well-being of people with psoriasis [202]. Thus, given the advantages of PBM, such as being a non-invasive therapy with few side effects and measurable benefits, treatment with this physical therapy deserves to be explored for the treatment of psoriasis. Thus, PBM could be presented as a promising therapy to alleviate symptoms and improve the quality of life of people with plaque psoriasis.

#### 4.3.2. Severe Atopic Dermatitis

Severe atopic dermatitis (SAD) is a common chronic inflammatory skin disease that predominantly affects children. However, it can persist into adulthood and/or begin at older ages. It is caused by numerous environmental factors, such as stress caused by various types of environmental pollution, immunological factors, including increased serum levels of immunoglobulin E (IgE) and imbalance between Th1 and Th2 type, as well as genetic factors. In SAD, the levels of IL-4, IL-6, and tumor necrosis factor α (TNF-α) tend to increase, while the level of IFN-γ tends to decrease. Furthermore, the number of Langerhans cells and the activation of mast cells increase in this pathology [203]. As for psoriasis, PBM has also been postulated as an effective complementary therapy, along with conventional therapies such as topical steroids or immunosuppressive medications for the treatment of SAD.

Leveraging the mechanisms of action of PBM, research in this area suggests that PBM may offer benefits in reducing inflammation, relieving symptoms, and improving the quality of life for people with SAD. Currently, the field of research is focused on determining optimal wavelengths and treatment protocols for PBM in SAD.

Ultraviolet radiation is frequently used as a second-line treatment for moderate to severe SAD in adults [204,205]. Its efficacy is based on the induction of T lymphocyte apoptosis, suppression of the antigen-presenting function of Langerhans cells and the production of anti-inflammatory mediators, as well as the ability to inhibit DNA synthesis and keratinocyte proliferation [206]. Itching is also another characteristic symptom of SAD. Given the anti-inflammatory and analgesic properties of PBM, this therapy could be applied to reduce itching and discomfort. On this matter, it has been described that PBM reduces the number of epidermal nerve fibers and the expression of axon guidance molecules, which is why it is also useful for the pruritus associated with SAD [207,208]. In addition, it has also been revealed that UV treatments are capable of modulating the immune response by positively regulating FoxP3-positive regulatory T cells [209]. Artificial light sources within the UV spectrum for the treatment of SAD range from broadband UVB (290–320 nm), narrow-band (NB) UVB (311–313 nm), excimer laser (308 nm), UVA-1 (340–400 nm), psoralens and UVA (PUVA) and combined UVA/UVB (280–400 nm), although medium-dose UVA1 and NB-UVB phototherapies have been reported to be the most effective modalities and safe for the treatment of SAD in adults [210,211]. UV can also be combined with the prior administration (oral or topical) of photosensitizing drugs such as psoralens (PDT). However, this is not considered the first modality of phototherapy treatment as it can present several side effects such as nausea, headache, fatigue, burning skin, itching and uneven skin pigmentation, as well as an increased risk of skin cancer In addition, it should also be noted that most patients prefer NB-UVB or UVA1 phototherapy, as they are easier to perform and do not require concomitant administration of a photosensitizer [212].

Recent work has shown that blue light induces an anti-inflammatory and antiproliferative effect in adult patients, which is why it may be beneficial for chronic inflammatory skin diseases such as SAD [213]. However, there is only one study in which patients with SAD have been treated with blue light. In this study, 36 hospitalized patients were treated five times a day with blue light (400 and 500 nm, 28.9 J/cm^2^) for a period of 6 months. At 15 days and at 3 and 6 months after starting the study, a decrease in the severity of the disease, in addition to itching, was observed between 29% and 54%. These patients also reported an improvement in sleep quality and an improvement in their well-being.

Red and near-infrared light can also be useful for the treatment of SAD. In this pathology, the hyperactivity of the immune system and inflammation play a central role; therefore, the anti-inflammatory effects can help modulate these immune responses and reduce the production of pro-inflammatory cytokines [202]. As in other autoimmune pathologies, the skin barrier function is impaired in SAD. Therefore, the application of this phototherapy may be of interest to regenerate the integrity of the skin barrier by promoting collagen synthesis. Clinical studies have revealed diverse information. In a double-blind clinical trial conducted with red LEDs and LED-NIR in patients with SAD, nine participants with grade II-III cellulite confirmed a reduction in cellulite after 3 months of follow-up [214]. In another case-control study, 28 patients were treated with red LED therapy (98 J/cm^2^ and 20 min/session). Patients treated with LEDs showed a recovery in half the time of those not treated, but 6 months after treatment, there were no significant differences between both groups [215].

Finally, in some cases and due to chronic scratching caused by the intense itching of SAD, some patients cause breaks in the skin that can lead to secondary infections. Patients with SAD have a decreased expression of antimicrobial peptides (AMPs), which facilitates dysbiotic colonization by Staphylococcus aureus, characteristic of the pathogenesis of this dermatitis [216]. Recently, the antimicrobial effect of blue light has been described in several research papers, showing how exposure to light in the range of 400–470 nm decreases viability in a heterogenous group of bacteria, including *Pseudomonas aeruginosa*, *Porphyromonas gingivalis*, *Helicobacter pylori* and methicillin-resistant *S. aureus* [57,217]. Therefore, using the antimicrobial properties of PBM could also reduce the risk of infection and promote overall skin health. It is critical to note that while PBM is promising, more well-designed clinical trials are needed to establish its efficacy, safety, and optimal long-term parameters for the treatment of severe atopic dermatitis.

## 5. Safety and Limitations of Photobiomodulation in Dermatology

PBM applied through LLLT or LED is generally regarded as safe. RL and NIR light do not seem to induce DNA damage, even at fluences up to 1280 J/cm^2^. In a clinical trial, intact skin was exposed for one minute at an irradiance of 1 W/cm^2^ with a low-power NIR laser with only transient erythema in one patient reported as adverse effects. In another study, skin temperature was measured after receiving a dose of 2 J/cm^2^ and compared to the control group without significant changes between groups. Concerning the potential for carcinogenesis or the application of this therapy to oncology patients, PBM has been widely utilized for alleviating skin and mucous membrane irritation post-radiotherapy for head and neck and breast cancers, with no reported pro-oncogenic effects [8,218,219]. A recently published paper stands out that, although there may be theoretical concerns, there is currently insufficient clinical evidence to substantiate the notion that PBM should be avoided in patients undergoing cancer treatment, those with a history of cancer, or individuals with cancer risk factors [21].

Noteworthy, existing evidence suggests that skin of color may be more susceptible to adverse effects from PBM wavelengths [3]. Visible light (400–700 nm) may produce non-transient hyperpigmentation in skin of color. A clinical trial demonstrated that skin of color presents higher photosensitivity to visible light, showing differences of up to 50% in the maximum tolerated dose of LED red light when compared to lighter skin types [220]. In this study, the authors defined the mentioned maximum tolerated dose as one that did not produce an adverse effect such as non-transient erythema or hyperpigmentation. In this regard, the use of lasers at higher doses (8–12 J/cm^2^) resulted in a significant increase in skin temperature and thermal-induced pain in skin color patients in another study. These effects may result from the absorption of photons by melanin in the epidermis of skin of color patients, which would raise the local temperature, subsequently leading to vasodilation, erythema, and pain [1,4].

Given these findings, it is advisable to be careful when adjusting PBM doses in the skin of color patients, initiating with safer parameters and scouting for adverse effects occurrence, looking for the maximum tolerated dose or a dose below it. It is remarkable that more evidence of PBM in darker skin phototypes is necessary since existing evidence on these subjects is limited.

As explained before, LED-light irradiation with wavelengths between 630 and 940 nm does not seem to produce severe harmful effects on humans. However, blue LED light (wavelengths of 400–500 nm) is not exempt from risk and published evidence suggests that blue light has the potential to cause injury to the skin, eyes, and some other human tissues [221,222,223].

Firstly, excessive exposure to blue LED light poses a potential risk to retinal function. Blue LED light can harm the retina in three main ways: thermally, mechanically, or photochemically. Mechanical damage typically occurs due to shock waves or sonic transients during short exposures and at high levels of irradiance. Thermal damage can result from temperature increases after absorbing enough blue LED light. Photochemical damage is caused by chemical reactions initiated by light [224].

In recent years, LED-light phototherapy has gained traction within the fields of dermatology and beauty for addressing skin conditions like psoriasis and acne. Despite its widespread use, concerns have emerged regarding the potential risks associated with a wide range of wavelengths spanning from violet to blue (380–440 nm). This spectrum has been linked to cellular damage and the potential formation of cancer through subsequent mutations [225].

As a result, exposure to blue-violet LED light is now deemed hazardous for treating skin diseases. Studies have shown that blue light, particularly when delivered at high irradiances, exhibits toxic effects in a manner dependent on both wavelength and dose. Specifically, exposure to wavelengths within the violet spectrum (400–440 nm) has been found to induce intracellular oxidative stress and cytotoxicity. Singlet oxygen is identified as a primary agent responsible for these harmful effects [226].

Addressing this challenge, the identification of a suitable range of fluences that effectively treat skin conditions without causing adverse side effects has become paramount when using blue light therapy.

Regarding PBM limitations, it is important to highlight that the majority of the existing evidence on the efficacy of its multiple applications lacks well-conducted randomized clinical trials and homogeneity on parameters or reproducibility, which, ultimately, makes it impossible to establish standard criteria for the particular laser, dosage, number, frequency, or duration of treatments on different skin diseases. Existing evidence against the efficacy of PBM in dermatology is scarce, probably because of the publication bias. These few studies do not recommend PBM application as they fail to demonstrate improvement in efficacy when compared to placebo, not because of the appearance of adverse effects in the treatment group [160,227,228,229,230].

## 6. Ethical Considerations

The effectiveness of PBM in terms of outcomes and clinical trials is not particularly remarkable, and the overall impression is that moderate results are achieved with repeated sessions. However, PBM is a safe treatment that can be considered as an adjunct to other therapies. On the other hand, LED devices are inexpensive and have potential in the development of devices that facilitate treatment compliance, such as home devices. Physicians can rely on these techniques with honest expectations explained to patients. Regulation varies across markets when acquiring those LED devices, and emphasis should be placed on LED devices marked by relevant authorities, as they are relatively easy to manufacture. Considerations should include certification, as only some LED devices are certified for medical purposes, the manufacturing company, wavelength, and the applied energy [231].

## 7. Conclusions

Photobiomodulation and its application in dermatology constitute a subject of discussion and recently raising interest; as a consequence, existing scientific evidence (well-structured, placebo-controlled, clinical trials) on the matter is still scarce, and a critical review of the literature does not allow to develop solid specific clinical recommendations on concrete applications. This is also difficult because PBM can be conducted in multiple ways and with an almost infinite combination of parameters, which hinders comparing the methods and results between published studies. However, we believe the value of the information offered in this review is increased when considering the mentioned recent nature of this therapy since it can help in the task of both identifying further indications for PBM and designing future studies that shed light on the picture.

In the authors’ opinion, one of the main weaknesses of the available data is that the impact of the cellular mechanisms caused by PBM has yet to be unraveled. It is the case of acute or chronic wounds that serve as an example to give an idea of the complexity of the objectives still to be considered even today. The finest cellular mechanisms are unknown, and numerous authors’ aim is still to identify a cellular reference point, such as effector cells, ignoring the fact that knowledge of the skin microenvironment is the key to fully understanding this process. To this, at least as regards the knowledge of the cellular mechanisms induced by therapies during wound healing, we must add the knowledge of the intricate interaction between the neurological system and the immune system since these are factors that have significant potential to facilitate the resolution during the treatment of any therapeutic interventions.

One aspect not discussed in this review is the ultraviolet radiation (UVR) potential as a photobiomodulation agent; this wavelength is better recognized for its detrimental effects, such as carcinogenesis or skin aging, but may also show a role in the modulation of homeostasis since skin exposure to UVR can trigger local responses secondary to the induction of chemical, hormonal, immune, and neural signals such as discussed recently by Slominski et al. [232].

PBM constitutes a promising and safe therapeutic tool in the management of multiple skin diseases. As stated previously, future efforts both from dermatologists and basic researchers should aim to clarify its mechanism of action and establish specific parameters and recommendations to be adopted during physicians’ daily routines, thus allowing for better clinical care for patients and the benefits of this encouraging therapy.

## Figures and Tables

**Figure 1 ijms-25-04483-f001:**
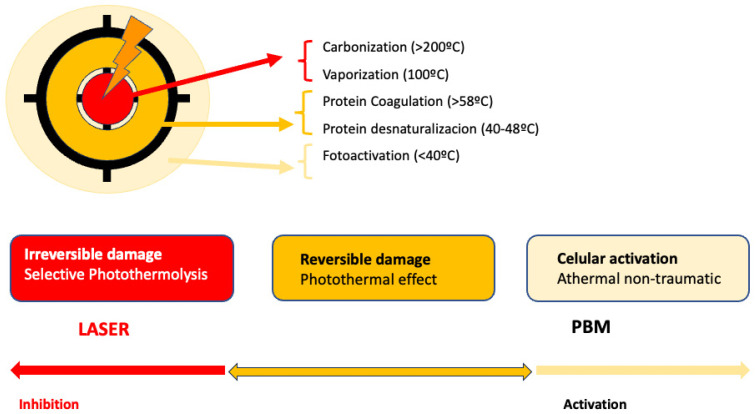
Represents the action of lasers in dermatology: In the center, selective photothermolysis occurs with the selective destruction of the chromophore and necrosis of the target tissue. In the surrounding tissue, PMB effects occur, corresponding to the dissipation of laser energy or lower doses.

**Figure 2 ijms-25-04483-f002:**
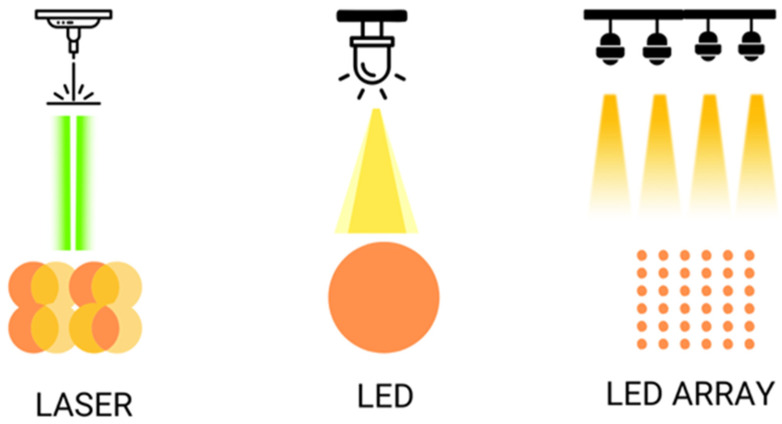
Shows the different ways of illumination with laser and LED, simple or on arrays.

**Figure 3 ijms-25-04483-f003:**
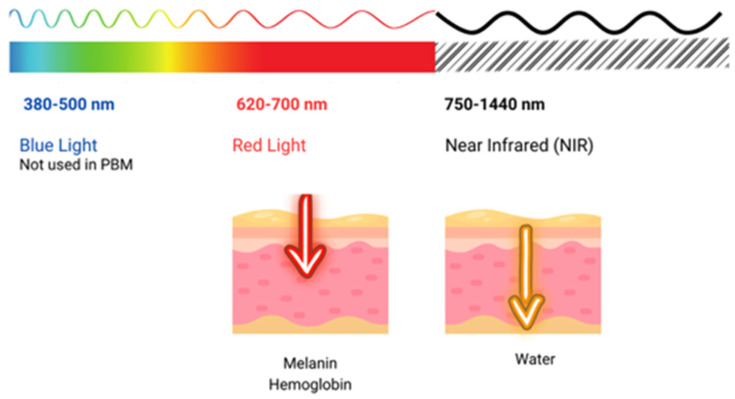
Shows the penetration of light at different wavelengths throughout the visible and near-infrared (NIR) spectrum.

**Table 1 ijms-25-04483-t001:** Summary of published clinical trials on PMB and photorejuvenation and wrinkles treatment.

Author/Year	Type of LED	Patients	Design of the Study	Protocol of Treatment	Results
Weis 2005 [95]	RL 590 nm	N = 90	8 treatments in 4 weeks 6 months follow-up	0.1 J/cm^2^ pulsing	90% of patients reduced photoaging signs. Histological response: - 90% improve Collagen I - 4% decrease MMPI
Russell 2005 [96]	RL 630 nm + NIR 830 nm	N = 31	9 light treatments Flow up weeks 9 and 12	RL 126 J/cm^2^ NIR 66 J/cm^2^	52% of patients reduced photoaging signs 81% of patients reported improvement in periocular wrinkles
Goldberg 2007 [97]	RL 630 nm + NIR 830 nm	N = 36	9 treatments in 12 weeks	RL 126 J/cm^2^ NIR 66 J/cm^2^	Significant improvement in softness, smoothness, and firmness
Yoon-Lee 2007 [98]	RL 630 nm + NIR 830 nm	N = 112	4 Groups: NIR, RL, NIR + RL and placebo 8 sessions, 4 weeks, and 3 months follow-up	RL 126 J/cm^2^ NIR 66 J/cm^2^	Both RL and NIR had effective and significant wrinkle reduction Skin elasticity better NIR and NIR + RL Melanin decrease RL
Baez 2007 [99]	RL 630 nm + NIR 830 nm	N = 30	9 sessions, 12 weeks	RL 126 J/cm^2^ NIR 66 J/cm^2^	91% color improvement 82% smoothness improvement 25–50% investigator assessment improvement
Wunsch 2014 [100]	RLT 611–650 nm ELT 570–850 nm	N = 136	2 sessions per week 30 treatments 3 Groups: RLT, ELT, and placebo		No difference between wavelengths Both treatments have significant differences in wrinkles
Hee-Nam 2017 [101]	RL 660 nm LED 411–777 nm	N = 52	1 session/day 12 weeks 2 Groups: RL, LED	5.17 J/cm^2^	Both treatments significantly improve wrinkles
Rocha-Mota 2023 [102]	RL 660 nm AL 590 nm	N = 137 Split-face	10 sessions periocular 4 weeks	3.8 J/cm^2^	Significant periocular wrinkles, with RL 31.6% and 29.9% with AL.

## Data Availability

All data are available in the manuscript.

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
