# Peer review of "Unlocking the Power of Light on the Skin: A Comprehensive Review on Photobiomodulation"

_ijms, 2024, doi:10.3390/ijms25084483_

Round 1

Reviewer 1 Report

Comments and Suggestions for Authors

Manuscript entitled, “Unlocking the Power of Light on the Skin: A Comprehensive Review on Photobiomodulation”, is an interesting review related to the exploration of clinical application of the photobiomodulation by using various light sources as an alternative therapy for various skin diseases. Although the manuscript is well written still, I’m having certain comments for enhanced readership in audience working in the closely related areas.

1.     The introduction section of the manuscript is too short and generalized and needs to be elaborated to put clear background for writing this particular review.

2.     In line 34-36 authors are saying that PBM is used to treat various skin disorders including scars, wounds, acne, ulcers etc. but in line 56-57 they are mentioning that “Therefore, progress of PBM development and its clinical applications must develop with close collaboration of basic research investigators”. These two sentences are contraindicatory with each other and needs to be justified or resolved.

3.     In section 2.2 it is not clear that which type and of what wavelength of light is utilized for Photodynamic Therapy.

4.     In table 1, results section of row 2 and 3 mentions 90% and 52% improvement in photoaging by using LED based PBM treatment, isn’t this statement is contraindicatory as how the use of a certain light can limit the photoaging effect caused in general. Needs to be justified.

5.     Conclusion section of the manuscript is too generalized and needs to be reframed by mentioning certain hypothesis generated out of the current review.

6.     Title of 6.1.1 should be reframed as “Development of Newer Therapeutics by Using Alternative Wavelengths”.

7.     The lights of certain wavelengths are also having harmful effects on our body including carcinogenicity and mutagenicity, these limitations should also be included in the review as a separate section.

8.     The manuscript needs to be rephrased to limit the text similarity and plagiarism.

Author Response

Montserrat Fernández-Guarino, MD, PhD.

Dermatology Department

Ramon y Cajal University Hospital, Madrid, Spain

Phone: (+34) 627654405

E-Mail: montsefernandezguarino@gmail.com

April 7th 2024

Dear Editor,

I would like to thank you for considering the publication of our paper in International Journal of Molecular Sciences. I’m summarizing here the changes incorporated following the reviewers’ suggestions:

Response to Reviewer 1 Comments:

Manuscript entitled, “Unlocking the Power of Light on the Skin: A Comprehensive Review on Photobiomodulation”, is an interesting review related to the exploration of clinical application of the photobiomodulation by using various light sources as an alternative therapy for various skin diseases. Although the manuscript is well written still, I’m having certain comments for enhanced readership in audience working in the closely related areas.

Point 1: The introduction section of the manuscript is too short and generalized and needs to be elaborated to put clear background for writing this particular review.

Response 1: We have re-elaborated and expanded the introduction, expanding the current background on photobiomodulation in dermatology and addressing in a more concrete way the objectives of the review and the method used.

Point 2: In line 34-36 authors are saying that PBM is used to treat various skin disorders including scars, wounds, acne, ulcers etc. but in line 56-57 they are mentioning that “Therefore, progress of PBM development and its clinical applications must develop with close collaboration of basic research investigators”. These two sentences are contraindicatory with each other and needs to be justified or resolved.

Response 2: The sentence in line 34-36 has been re-elaborated to avoid inconsistencies. We were referring to the development of new indications and new clinical protocols.

Point 3: In section 2.2 it is not clear that which type and of what wavelength of light is utilized for Photodynamic Therapy.

Response 3: We specified in concrete terms the type of PDT and wavelength used: “PBM through PDT for the treatment of wounds has been proven to be effective when using red light LED at 630 nm prior application of 20% ALA in liposomal gel as photosensitizer agent, reaching light doses near 80 J/cm2 per session and repeating the session weekly up to three times [64].”

Point 4:  In table 1, results section of row 2 and 3 mentions 90% and 52% improvement in photoaging by using LED based PBM treatment, isn’t this statement is contraindicatory as how the use of a certain light can limit the photoaging effect caused in general. Needs to be justified.

Response 4: We modified Table 1 in order to express in a clearer way the results of the studies referred to by the reviewer: the studies referred to any form of improvement or rejuvenation (reduction of signs of photoaging such as smoother texture, reduction of peri-orbital rhytids, and reduction of erythema and pigmentation by promoting dermal remodeling) was found in 90% and 52% of patients involved on the study, respectively.

Point 5: Conclusion section of the manuscript is too generalized and needs to be reframed by mentioning certain hypothesis generated out of the current review.

Response 5: We have rewritten the conclusion of the article, expanding it and making it more concrete according to the hypotheses extracted from the manuscript.

Point 6: Title of 6.1.1 should be reframed as “Development of Newer Therapeutics by Using Alternative Wavelengths”.

Response 6: We have reframed the title as suggested

Point 7:  The lights of certain wavelengths are also having harmful effects on our body including carcinogenicity and mutagenicity, these limitations should also be included in the review as a separate section.

Response 7: We have included a new section (5. Safety and Limitations of Photobiomodulation in Dermatology.) in which we discuss, based on recent literature, the aspects related to the limitations of photobiomodulation in dermatology, including adverse effects and precautions.

Point 8: The manuscript needs to be rephrased to limit the text similarity and plagiarism.

Response 8: We have re-elaborated the introduction and conclusion section, after that, we have examined the manuscript with the tool Turnitin which allows to detect plagiarism obtaining a very low percentage of similarity.

Response to Reviewer 2 Comments

I congratulate the authors of this study. Review works on this topic appear in the literature, e.g. Dompe C, Moncrieff L, Matys J, Grzech-Leśniak K, Kocherova I, Bryja A, Bruska M, Dominiak M, Mozdziak P, Skiba THI, Shibli JA, Angelova Volponi A, Kempisty B, Dyszkiewicz-Konwińska M. Photobiomodulation-Underlying Mechanism and Clinical Applications. J Clin Med. 2020 Jun 3;9(6):1724. doi: 10.3390/jcm9061724.In the introduction, the authors should mention review articles on this therapy that have been published in recent years and demonstrate how their review differs. Furthermore, I suggest the following

Point 1: Since the authors submit this article as a review, they should add in the introductory part and in the Abstract information about the period of time they took into account when collecting the literature, and what the literature search criteria were (inclusion/exclusion).

Response 1: We have re-elaborated and expanded the introduction, expanding the current background on photobiomodulation in dermatology (including recent reviews on the topic as suggested) and addressing in a more concrete way in the introduction and in the Abstract the objectives of the review as well as the methods used for the review.

Point 2: Chapters 4, 5, 6 should be in order. The conclusion should be the last chapter. Chapter titled 6. Future Directions is actually a continuation of the review and should be attached to the previous chapters.

Response 2: We have rearranged the chapters as suggested

Point 3: The paper describes the evidence for the effectiveness of PBM therapy. No information about studies questioning its effectiveness. Studies questioning the effectiveness of this method should also be included. Write in what cases it should not be used and when the effects are mediocre. Then chapter 4, i.e. Ethical and Safety Considerations, will be more understandable.

Response 3: We have included a new section (5. Safety and Limitations of Photobiomodulation in Dermatology.) in which we discuss, based on recent literature, the aspects related to the limitations of photobiomodulation in dermatology, including adverse effects and precautions as well as discussing the scarcity of solid evidence in certain indications

I look forward to hearing from you at your earliest convenience.

Yours sincerely,

Dr. Montserrat Fernandez-Guarino

Reviewer 2 Report

Comments and Suggestions for Authors

I congratulate the authors of this study. Review works on this topic appear in the literature, e.g. Dompe C, Moncrieff L, Matys J, Grzech-Leśniak K, Kocherova I, Bryja A, Bruska M, Dominiak M, Mozdziak P, Skiba THI, Shibli JA, Angelova Volponi A, Kempisty B, Dyszkiewicz-Konwińska M. Photobiomodulation-Underlying Mechanism and Clinical Applications. J Clin Med. 2020 Jun 3;9(6):1724. doi: 10.3390/jcm9061724.

In the introduction, the authors should mention review articles on this therapy that have been published in recent years and demonstrate how their review differs. Furthermore, I suggest the following:

1. Since the authors submit this article as a review, they should add in the introductory part and in the Abstract information about the period of time they took into account when collecting the literature, and what the literature search criteria were (inclusion/exclusion).

2. Chapters 4, 5, 6 should be in order. The conclusion should be the last chapter. Chapter titled 6. Future Directions is actually a continuation of the review and should be attached to the previous chapters.

3. The paper describes the evidence for the effectiveness of PBM therapy. No information about

studies questioning its effectiveness. Studies questioning the effectiveness of this method should also be included. Write in what cases it should not be used and when the effects are mediocre. Then chapter 4, i.e. Ethical and Safety Considerations, will be more understandable.

Author Response

(The authors gave the same response as above.)

Round 2

Reviewer 1 Report

Comments and Suggestions for Authors

The author's have resolved the queries raised by the reviewers and the manuscript is in good shape now to be accepted for publication

Author Response

Montserrat Fernández-Guarino, MD, PhD.

Dermatology Department

Ramon y Cajal University Hospital, Madrid, Spain

Phone: (+34) 627654405

E-Mail: montsefernandezguarino@gmail.com

April 15th 2024

Dear Editor,

I would like to thank you for considering the publication of our paper in International Journal of Molecular Sciences. I’m summarizing here the changes incorporated following your suggestion:

“This is an interesting paper that for the most part addressed the reviewers critique. I also reviewed it and believe that it would benefit from short mentioning in the limitation section on diverse functions of UVR as discussed in (Proceedings of the National Academy of Sciences. 2024;121(14):e2308374121. doi: doi:10.1073/pnas.2308374121) since similar functions could be triggered by longer wavelength of solar radiation”

Response:

We have incorporated a short paragraph on that topic in the limitations section as suggested: “One aspect not discussed in this review is the ultraviolet radiation (UVR) potential as a photobiomodulation agent, this wavelength is better recognized for its detrimental ef-fects such as carcinogenesis or skin aging but may also show a role in modulation of ho-meostasis since skin exposure to UVR can trigger local responses secondary to the induc-tion of chemical, hormonal, immune, and neural signals such as discussed recently by Slominski et al. [234].”

I look forward to hearing from you at your earliest convenience.

Yours sincerely,

Dr. Montserrat Fernandez-Guarino

Reviewer 2 Report

Comments and Suggestions for Authors

Congratulations to the authors. I have no objections. I recommend the article for publication in its current version.

Author Response

(The authors gave the same response as above.)
